# Genetic and pharmacological relationship between P-glycoprotein and increased cardiovascular risk associated with clarithromycin prescription: An epidemiological and genomic population-based cohort study in Scotland, UK

Ify R. Mordi [1]*, Benjamin K. Chan [2], N. David Yanez [2], Colin N. A. Palmer [3], Chim C. Lang[1], James D. Chalmers[1]*

**1** Division of Molecular and Clinical Medicine, University of Dundee, Dundee, United Kingdom, **2** School of Public Health, Oregon Health and Science University and Portland State University, Portland, Oregon, United States of America, **3** Division of Population Health and Genomics, University of Dundee, Dundee, United Kingdom

\* i.mordi@dundee.ac.uk (IRM); j.chalmers@dundee.ac.uk (JDC)

## Abstract

### Background

There are conflicting reports regarding the association of the macrolide antibiotic clarithromycin with cardiovascular (CV) events. A possible explanation may be that this risk is partly mediated through drug–drug interactions and only evident in at-risk populations. To the best of our knowledge, no studies have examined whether this association might be mediated via P-glycoprotein (P-gp), a major pathway for clarithromycin metabolism. The aim of this study was to examine CV risk following prescription of clarithromycin versus amoxicillin and in particular, the association with P-gp, a major pathway for clarithromycin metabolism.

### Methods and findings

We conducted an observational cohort study of patients prescribed clarithromycin or amoxicillin in the community in Tayside, Scotland (population approximately 400,000) between 1 January 2004 and 31 December 2014 and a genomic observational cohort study evaluating genotyped patients from the Genetics of Diabetes Audit and Research Tayside Scotland (GoDARTS) study, a longitudinal cohort study of 18,306 individuals with and without type 2 diabetes recruited between 1 December 1988 and 31 December 2015. Two single-nucleotide polymorphisms associated with P-gp activity were evaluated (rs1045642 and rs1128503 —AA genotype associated with lowest P-gp activity). The primary outcome for both analyses was CV hospitalization following prescription of clarithromycin versus amoxicillin at 0–14 days, 15–30 days, and 30 days to 1 year. In the observational cohort study, we calculated hazard ratios (HRs) adjusted for likelihood of receiving clarithromycin using

**Data Availability Statement:** De-identified data for both the longitudinal cohort and pharmacogenomic parts of the study are held within the Tayside Health Informatics Centre Safehaven (https://www.dundee.ac.uk/hic/hicsafehaven/). Applications for access to the GoDARTS dataset can be made via a Project Collaboration Request Form (https://godarts.org/scientific-community/). Code used in the construction of the dataset and statistical analysis is available at the following link: https://github.com/ifymordi/Clarithromycin.

**Funding:** Funding: This study was supported by a research grant from Tenovus Scotland and the Jimmie Cairncross Charitable Trust. IRM is supported by National Health Service Education for Scotland/Chief Scientist Office Postdoctoral Clinical Lectureship (PCL 17/07). The GoDARTS study was supported by the following: genotyping was facilitated by capital funding from the Scottish Government Chief Scientist Office Generation Scotland initiative (www.generationscotland.org); The Wellcome Trust U.K. type 2 diabetes case control collection (GoDARTS2) was funded by a Wellcome Trust [grant number GR02960] and the GWAS genotyping was performed as part of the Wellcome Trust Case Control Consortium 2 [084726/Z/08/Z, 085475/Z/08/Z, 085475/B/08/Z]. JDC is supported by the British Lung Foundation Chair of Respiratory Research. Role of the Funding Source The funder of the study had no role in study design, data collection, data analysis, data interpretation, or writing of the report.

**Competing interests:** The authors have declared that no competing interests exist.

**Abbreviations:** CLARICOR, Effect of Clarithromycin on Mortality and Morbidity in Patients With Ischemic Heart Disease Trial; COPD, chronic obstructive pulmonary disease; CV, cardiovascular; CYP3A4, cytochrome p4503A4; FDA, United States Food and Drug Administration; HR, hazard ratio; MI, myocardial infarction; P-gp, permeability-glycoprotein; SMD, standardized mean difference; SNP, single-nucleotide polymorphism.

inverse proportion of treatment weighting as a covariate, whereas in the pharmacogenomic study, HRs were adjusted for age, sex, history of myocardial infarction, and history of chronic obstructive pulmonary disease.

The observational cohort study included 48,026 individuals with 205,227 discrete antibiotic prescribing episodes (34,074 clarithromycin, mean age 73 years, 42% male; 171,153 amoxicillin, mean age 74 years, 45% male). Clarithromycin use was significantly associated with increased risk of CV hospitalization compared with amoxicillin at both 0–14 days (HR 1.31; 95% CI 1.17–1.46, $p < 0.001$) and 30 days to 1 year (HR 1.13; 95% CI 1.06–1.19, $p < 0.001$), with the association at 0–14 days modified by use of P-gp inhibitors or substrates (interaction $p$-value: 0.029). In the pharmacogenomic study (13,544 individuals with 44,618 discrete prescribing episodes [37,497 amoxicillin, mean age 63 years, 56% male; 7,121 clarithromycin, mean age 66 years, 47% male]), when prescribed clarithromycin, individuals with genetically determined lower P-gp activity had a significantly increased risk of CV hospitalization at 30 days to 1 year compared with heterozygotes or those homozygous for the non-P-gp–lowering allele (rs1045642 AA: HR 1.39, 95% CI 1.20–1.60, $p < 0.001$, GG/GA: HR 0.99, 95% CI 0.89–1.10, $p = 0.85$, interaction $p$-value < 0.001 and rs1128503 AA 1.41, 95% CI 1.18–1.70, $p < 0.001$, GG/GA: HR 1.04, 95% CI 0.95–1.14, $p = 0.43$, interaction $p$-value < 0.001). The main limitation of our study is its observational nature, meaning that we are unable to definitively determine causality.

## Conclusions

In this study, we observed that the increased risk of CV events with clarithromycin compared with amoxicillin was associated with an interaction with P-glycoprotein.

---

## Author summary

### Why was this study done?

- Macrolide antibiotics such as clarithromycin are frequently used for treatment of lower respiratory tract infections.

- Several studies have reported increased cardiovascular risk associated with clarithromycin prescription, including the Effect of Clarithromycin on Mortality and Morbidity in Patients With Ischemic Heart Disease (CLARICOR) randomized-controlled trial.

- The mechanism behind this increased risk is unclear; however, it would be important to identify patients who might be at higher risk when prescribed clarithromycin in whom alternative antibiotics such as amoxicillin could be used.

### What did the researchers do and find?

- We conducted an observational study in 2 parts to examine the association between clarithromycin prescription and cardiovascular outcome.

- First, we performed a propensity-weighted observational population cohort analysis comparing clarithromycin versus amoxicillin prescription and cardiovascular outcome in patients in Tayside, Scotland.

- In this analysis, we found that that patients prescribed clarithromycin were significantly more likely to have a cardiovascular hospitalization at 0–14 days and 30 days to 1 year after prescription than those prescribed amoxicillin and that individuals who were co-prescribed P-glycoprotein substrates or inhibitors and clarithromycin had a significantly higher risk of cardiovascular hospitalization.

- To explore these findings further, we performed a pharmacogenomic study involving patients with available genotype data from the Genetics of Diabetes Audit and Research Tayside Scotland (GoDARTS) study.

- Using 2 genetic variants associated with P-glycoprotein activity (rs1045642 and rs1128503), we found that patients with the AA genotype, which for both variants is associated with reduced P-glycoprotein activity, had a significantly increased risk of cardiovascular hospitalization between 30 days and 1 year.

**What do these findings mean?**

- Our findings raise the possibility that the observed association between clarithromycin prescription and increased cardiovascular risk could be mediated by P-glycoprotein.

- These results may have implications for clarithromycin use in patients taking P-glycoprotein inhibitors or with low genetically predicted P-glycoprotein activity.

## Introduction

Clarithromycin is a widely prescribed macrolide antibiotic, comprising around 15% of all primary care antibiotic prescriptions in the United Kingdom, recommended for treatment of patients with lower respiratory tract infections either as monotherapy or in combination [1–3]. There has been growing concern regarding increased cardiovascular (CV) risk of clarithromycin. The Effect of Clarithromycin on Mortality and Morbidity in Patients With Ischemic Heart Disease (CLARICOR) trial in patients with high CV risk [4], designed to test the hypothesis that clarithromycin would reduce CV risk, actually found that 2 weeks of clarithromycin caused a 45% relative risk increase in CV mortality compared with placebo. These results were supported by a number of observational studies [5–8] and meta-analyses [9, 10] suggesting that clarithromycin and other macrolide antibiotics were associated with adverse outcome, not only in the short term during and after exposure, but also in the longer term after drug discontinuation, leading to a recent United States Food and Drug Administration (FDA) safety alert on the use of clarithromycin in patients with heart disease [11, 12]. Nonetheless, recently, studies in lower-risk community populations have not found this association. An emerging hypothesis is that these conflicting results suggest that the high CV risk with clarithromycin may only be present in a subset of individuals [13–16].

A potential risk modifier is concurrent medication use. Clarithromycin is both a substrate and inhibitor of both cytochrome p4503A4 (CYP3A4) [17] and permeability-glycoprotein (P-gp) [18], meaning that circulating clarithromycin levels are affected by alterations in CYP3A4

or P-gp activity [19]. Several studies have investigated the possibility of an interaction between clarithromycin, concomitant CYP3A4 medication use, and CV risk; however, no robust association has been found [6, 14, 20]. To the best of our knowledge, there have been no original research studies evaluating whether the association between clarithromycin use and CV risk is modulated via P-gp. Both animal [19] and clinical studies [21, 22] suggest that co-administration of P-gp inhibitors such as verapamil, omeprazole, nelfinavir [23], or ketoconazole with macrolide antibiotics leads to an increase in the oral bioavailability of macrolides and increased plasma levels. Further support for an interaction between P-gp inhibition and macrolides comes from genetic studies showing that patients with genetic variants associated with low P-gp activity also have higher levels of macrolides when exposed to these antibiotics [24]. The use of pharmacogenomics may help overcome some of the limitations of observational studies and help shed light on potential drug interactions.

We hypothesized that the increased CV risk associated with clarithromycin may be linked with concurrent use of P-gp inhibitors or substrates and that individuals with genotypes associated with low P-gp activity, a proxy for P-gp inhibition, would also have an increased CV risk when prescribed clarithromycin.

## Methods

The study consisted of 2 parts: a traditional observational cohort study and a pharmacogenomic study.

## Observational cohort study

The prospective analysis plan for this section of the study can be found in S1 Text. The study population was all approximately 400,000 residents of the Tayside region of Scotland registered with an NHS Tayside general practice at any point in the study period (2004–2014). Demographic and community prescribing data were obtained through the Health Informatics Centre (HIC), University of Dundee, which provides anonymized linked individual patient data, including prescribing of antibiotics as previously described [25]. These datasets were linked to other datasets including demographic, clinical, hospital admission, and mortality data that are linked by a unique 10-digit patient identifier (the Community Health Index number) used for all healthcare activities in Scotland. All research data are robustly anonymized and approved by the Tayside National Health Service Caldicott Guardian under an overarching ethical approval for anonymized data research using HIC. Prescribing data between 2004 and 2014 were used to identify all patients over 18 years old who were prescribed clarithromycin (alone or in combination with amoxicillin or another antibiotic) over this period. A control group of individuals prescribed amoxicillin as a sole antibiotic was also identified. A propensity-score model for clarithromycin exposure was estimated using baseline demographic and clinical covariates hypothesized to be relevant shown in Table 1. Outcome models were weighted using inverse propensity of treatment weights (IPTWs) to account for baseline differences between exposure cohorts and robust sandwich covariance estimation to account for multiple exposures for each individual.

## Pharmacogenomic cohort study: Association of genetically determined P-gp activity, clarithromycin, and CV mortality

We did not have a prespecified analysis plan for the pharmacogenomic study; however, the analysis was informed by the observational cohort study. Patients with available prescribing data were obtained from the Genetics of Diabetes Audit and Research in Tayside Scotland study (GoDARTS), the details of which have been published previously [26]. In brief, this is a

**Table 1. Baseline characteristics of observational cohort study.**

| | Clarithromycin | Amoxicillin | Standardized Difference |
|---|---|---|---|
| Total Number of Unique Patients | 11,489 | 36,537 | |
| Total Number of Prescriptions | 34,074 | 171,153 | |
| Age at Prescription (years) [mean ± SD] | 73.3 ± 12.3 | 74.2 ± 13.2 | 0.070 |
| Male | 14,280 (41.9) | 76,521 (44.7) | 0.056 |
| Type 2 Diabetes | 5,555 (16.3) | 27,942 (16.3) | <0.001 |
| Chronic Obstructive Pulmonary Disease | 8,647 (25.4) | 25,282 (14.8) | 0.267 |
| Prior Myocardial Infarction | 1,289 (3.8) | 7,888 (4.6) | 0.040 |
| Prior Heart Failure | 1,111 (3.3) | 6,433 (3.8) | 0.027 |
| Clinically Indicated Echocardiography within previous year | 3,151 (9.2) | 13,913 (8.1) | 0.039 |
| History of Left Ventricular Systolic Impairment | 359 (1.1) | 1,743 (1.0) | 0.010 |
| Angiotensin Converting Enzyme Inhibitor | 11,583 (34.0) | 62,533 (36.5) | 0.052 |
| Angiotensin II Receptor Blocker | 167 (0.5) | 950 (0.6) | 0.014 |
| Aspirin | 14,660 (43.0) | 79,532 (46.4) | 0.068 |
| Beta Blocker | 7,505 (22.0) | 48,197 (28.2) | 0.141 |
| Clopidogrel | 3,621 (10.6) | 17,316 (10.1) | 0.016 |
| Dihydropyridine Calcium Channel Blocker | 8,661 (25.4) | 47,993 (28.0) | 0.059 |
| Loop Diuretic | 15,292 (44.9) | 66,654 (38.9) | 0.122 |
| Mineralocorticoid Receptor Antagonist | 3,099 (9.1) | 14,140 (8.3) | 0.028 |
| Nondihydropyridine Calcium Channel Blocker | 527 (1.5) | 2,547 (1.5) | <0.001 |
| Statin | 14,105 (41.4) | 76,308 (44.6) | 0.065 |
| Thiazide Diuretic | 7,678 (22.5) | 42,877 (25.0) | 0.059 |
| Warfarin | 3,620 (10.6) | 20,217 (11.8) | 0.038 |
| CYP3A4 inhibitor/substrate | 7,902 (23.2) | 31,664 (18.5) | 0.116 |
| P-glycoprotein inhibitor/substrate | 15,080 (44.2) | 78,285 (45.7) | 0.030 |
| Nonsteroidal anti-inflammatory drug | 21,973 (64.5) | 112,831 (65.9) | 0.029 |

Figures represent mean ± SD or number with percentage in parentheses. Percentages are reported as a proportion of the number of prescriptions. CYP3A4, cytochrome p4503A4.

longitudinal cohort study comprising 18,306 individuals, 10,149 with type 2 diabetes (T2D) and 8,157 controls without T2D at the time of recruitment, of which genotype data were available for 8,564 T2D individuals and 4,586 controls. Genotyping data have been previously described in full [26]. A blood sample for genotyping was obtained from individuals at baseline, and all patients consented to electronic record linkage, allowing details on prescriptions from 1989 to present and outcome data on deaths and hospitalizations. In this part of the study, we again only included individuals who had received a prescription for either amoxicillin or clarithromycin. Collection and analysis of data in GoDARTS was approved by the East of Scotland Research and Ethics Committee. All participants had given written consent for their data to be linked and analyzed for research purposes.

We selected 2 single-nucleotide polymorphisms (SNPs) within the human multidrug-resistance MDR1 gene (ABCB1), which codes for P-gp, which have been shown to be associated with P-gp activity in white healthy volunteers and for which there was a reasonably high prevalence of each genotype—rs1045642, rs1128503 [27–31]. Patients were categorized into 2 groups based on genotype with those individuals homozygous for the risk allele (the allele associated with reduced P-gp activity—A for both SNPs), i.e., individuals with low genetically predicted P-gp activity compared with heterozygous individuals and those homozygous for the nonrisk allele (i.e., intermediate and high genetically determined P-gp activity).

## Study endpoints

The primary endpoint for both studies was CV hospitalization. In our initial funding proposal, we planned to evaluate CV mortality as the primary endpoint; however, we felt we would be likely to be underpowered based on the number of patients and changed this to CV hospitalization before the preliminary analysis. Following the date of prescription, patients were followed up for CV hospitalizations, myocardial infarction, and death based on International Classification of Diseases (ICD)-10 coding for 1 year. The following codes were used: CV hospitalization or death –I00-I99; myocardial infarction—I21, I22, and I23. As previous studies have suggested that the CV risk may be different in the short term versus longer time periods, we stratified outcomes at 0–14 days, 15–30 days, and 30 days to 1 year post-prescription. Following the 1-year period of follow-up post-prescription, or after a censored event, surviving patients could be re-entered into the study if they had a further prescription of either antibiotic, similar to the methodology used in other large studies of this type [8, 32].

## Statistical analysis

Continuous variables are reported as mean ± standard deviation, and categorical variables are reported as number and percentage. Standardized mean differences (SMDs) between the clarithromycin and amoxicillin groups are reported, with an SMD > 0.1 considered a significant difference between the groups.

In the observational cohort study, Cox proportional hazards regression was performed for the outcome of CV hospitalization or hospitalization for MI at 0–14 days, 15–30 days, and 30 days to 1 year. Additionally, we evaluated the endpoints of all-cause and CV mortality. Adjusted HRs in the observational study are reported adjusting for the IPTW, using the propensity score for likelihood of prescription of clarithromycin based on baseline variables reported in Table 1 [33]. Analysis for the pharmacogenomic study was informed by the design and results of the observational study. In the pharmacogenomic study, a multivariable Cox regression analysis was performed for the association between clarithromycin use versus amoxicillin on CV hospitalization with adjustment for age at the time of antibiotic prescription, sex, history of prior myocardial infarction (MI), chronic obstructive pulmonary disease (COPD), and T2D. Interaction testing was performed to determine whether there was a significant difference between amoxicillin and clarithromycin prescribing depending on P-gp activity. All tests were 2-sided. Statistical analysis was performed using SAS (version 9.4, SAS Institute, https://www.sas.com/en_gb/software/stat.html), STATA (version 14.0, StataCorp, https://www.stata.com/), and R (version 3.5.1, The R Project for Statistical Computing, https://www.r-project.org/).

This study is reported as per the Strengthening the Reporting of Observational Studies in Epidemiology (STROBE) guideline (S1 Checklist).

# Results

## Observational longitudinal cohort study

**Baseline characteristics.** Over the duration of the study, there were 34,074 prescriptions for clarithromycin for 11,489 unique individuals and 171,153 amoxicillin prescriptions for 36,537 unique individuals. The mean age at prescription was 73.3 ± 12.3 (mean ± standard deviation) years in the clarithromycin group, and 41.9% were male, whereas in the amoxicillin group, the mean age was 74.2 ± 13.2 years, and 44.7% were male. Baseline characteristics are summarized in Table 1. Overall, individuals prescribed clarithromycin more likely to have

**Table 2. Clinical outcomes in the observational cohort study (unadjusted counts and percentages).**

| | | Clarithromycin (*n* = 34,074) | Amoxicillin (*n* = 171,153) |
|---|---|---|---|
| **Cardiovascular Hospitalization** | 0–14 Days | 559 (1.6) | 2,355 (1.4) |
| | 15–30 Days | 431 (1.3) | 2,105 (1.2) |
| | 30 Days–1 Year | 1,828 (5.4) | 11,321 (6.6) |
| **Hospitalization for MI** | 0–14 Days | 34 (0.1) | 169 (0.1) |
| | 15–30 Days | 23 (0.07) | 131 (0.08) |
| | 30 Days–1 Year | 164 (0.5) | 1,076 (0.6) |
| **Cardiovascular Mortality** | 0–14 Days | 73 (0.2) | 532 (0.3) |
| | 15–30 Days | 69 (0.2) | 558 (0.3) |
| | 30 Days–1 Year | 508 (1.5) | 3,925 (2.3) |
| **All-Cause Mortality** | 0–14 Days | 289 (0.8) | 1,601 (1.0) |
| | 15–30 Days | 277 (0.8) | 1,722 (1.0) |
| | 30 days–1 Year | 1,530 (4.5) | 9,991 (5.8) |

Figures in parentheses refer to the number of events as a percentage of the total number of prescriptions. MI, myocardial infarction

COPD and be prescribed loop diuretics and CYP3A4 inhibitors or substrates and were less likely to be taking beta-blockers (SMD > 0.1).

**Clinical Outcomes.** Clinical outcomes of CV hospitalization, hospitalization for MI, CV mortality, and all-cause mortality at 0–14 days, 15–30 days, and 30 days to 1 year are summarized in Table 2. Within the first 14 days following clarithromycin prescription, there were 559 CV hospitalizations (1.6% of all prescriptions) and 289 deaths from any cause, compared with the amoxicillin group in which there were 2,355 CV hospitalizations (1.2% of all prescriptions) and 1,601 deaths from any cause.

In our propensity-weighted analysis, despite there being a higher number of absolute events in the amoxicillin group, clarithromycin use was significantly associated with an increased risk of CV hospitalization compared with amoxicillin only at both 0–14 days (unadjusted absolute risk 1.6% versus 1.4%; propensity-weighted hazard ratio (HR) 1.31; 95% CI 1.17–1.46, $p < 0.001$) and 30 days to 1 year (unadjusted absolute risk 1.3% versus 1.2%; propensity-weighted HR 1.13; 95% CI 1.06–1.19, $p < 0.001$) but not at 15–30 days (unadjusted absolute risk 5.4% versus 6.6%; propensity-weighted HR 1.11; 95% CI 0.98–1.26, $p = 0.09$) (Table 3). A

**Table 3. Association of clarithromycin and cardiovascular risk versus amoxicillin in the observational study.**

| | 0–14 Days | | | 15–30 Days | | | 30 Days–1 Year | | |
|---|---|---|---|---|---|---|---|---|---|
| Outcome | Crude Hazard Ratio (95% CI) | Adjusted Hazard Ratio (95% CI) | *p*-value | Crude Hazard Ratio (95% CI) | Adjusted Hazard Ratio (95% CI) | *p*-value | Crude Hazard Ratio (95% CI) | Adjusted Hazard Ratio (95% CI) | *p*-value |
| Cardiovascular Hospitalization | 1.22 (1.10–1.34) | 1.31 (1.17–1.46) | <0.001 | 1.08 (0.97–1.21) | 1.11 (0.98–1.26) | 0.09 | 1.05 (1.00–1.11) | 1.13 (1.06–1.19) | <0.001 |
| Hospitalization for Myocardial Infarction | 1.04 (0.72–1.49) | 1.37 (0.89–2.11) | 0.16 | 0.89 (0.57–1.40) | 0.94 (0.54–1.62) | 0.82 | 1.00 (0.84–1.18) | 1.02 (0.85–1.24) | 0.82 |
| Cardiovascular Mortality | 0.85 (0.77–0.93) | 0.93 (0.81–1.06) | 0.25 | 0.66 (0.51–0.85) | 0.82 (0.62–1.10) | 0.18 | 0.85 (0.71–1.01) | 0.96 (0.87–1.07) | 0.46 |
| All-Cause Mortality | 0.86 (0.77–0.96) | 1.05 (0.92–1.20) | 0.43 | 0.80 (0.71–0.90) | 0.93 (0.81–1.06) | 0.25 | 0.95 (0.91–1.00) | 0.97 (0.92–1.02) | 0.18 |

*p*-values were estimated using robust covariance sandwich estimation and refer to the adjusted hazard ratio, which was adjusted using the likelihood of clarithromycin prescription as a covariate (inverse probability of treatment weighting). This included the following variables: age at prescription, sex, prior history of chronic obstructive pulmonary disease, prior myocardial infarction, history of type 2 diabetes, left ventricular systolic function impairment, and all medications listed in Table 1. CI, confidence interval

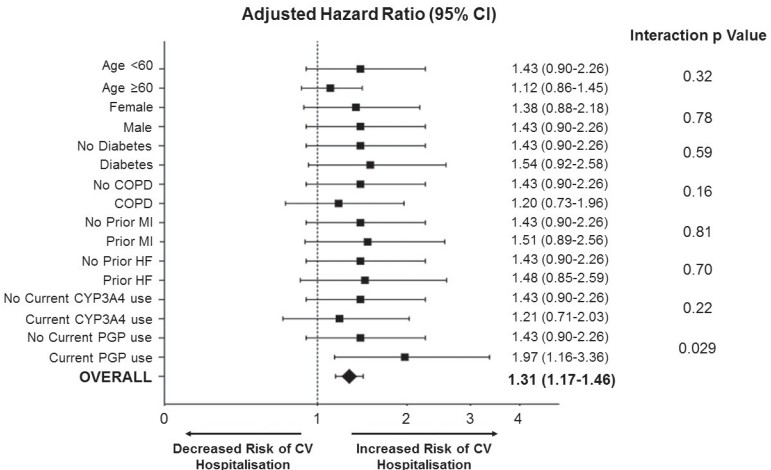

**Fig 1. Subgroup analysis of risk of CV hospitalization at 14 days associated with clarithromycin use versus amoxicillin.** Hazard ratio adjusted using the likelihood of clarithromycin prescription as a covariate (inverse probability of treatment weighting)—this included the following variables: age at prescription, sex, prior history of COPD, prior MI, history of type 2 diabetes, left ventricular systolic function impairment, and all medications listed in Table 1. CI, confidence interval; COPD, chronic obstructive pulmonary disease; CV, cardiovascular; CYP3A4, cytochrome P450 3A4; HF, heart failure; MI, myocardial infarction; PGP, P-glycoprotein.

higher proportion of those taking clarithromycin had MI requiring hospitalization within 14 days, though this difference was not statistically significant (adjusted HR 1.37; 95% CI 0.89–2.11, $p$ = 0.12). There was no significant difference in all-cause or CV mortality between clarithromycin and amoxicillin at any time point.

When stratified by baseline clinical characteristics, the only significant interaction with clarithromycin use and the primary outcome was in patients with a concomitant prescription for medications metabolized through P-gp (Figs 1 and 2, unadjusted results in S1 Table). Patients prescribed clarithromycin who were also taking P-gp inhibitors were significantly more likely to have a CV hospitalization within the first 14 days (HR 1.97; 95% CI 1.16–3.36),

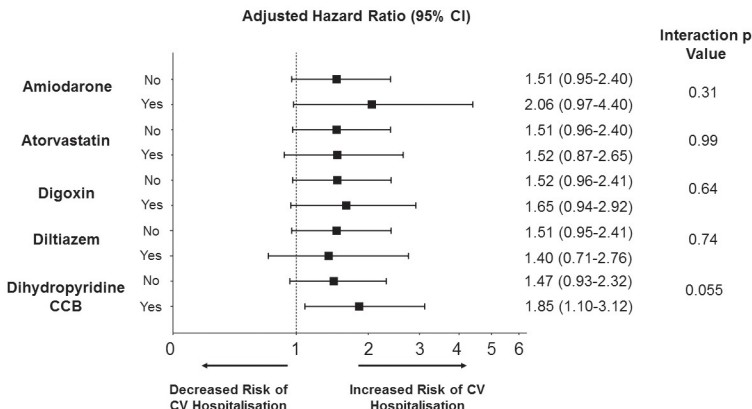

**Fig 2. Risk of CV hospitalization at 14 days stratified by concomitant P-glycoprotein medication prescription.** Hazard ratio adjusted using the likelihood of clarithromycin prescription as a covariate (inverse probability of treatment weighting)—this included the following variables: age at prescription, sex, prior history of chronic obstructive pulmonary disease, prior myocardial infarction, history of type 2 diabetes, left ventricular systolic function impairment, and all medications listed in Table 1. CCB, calcium channel blocker; CI, confidence interval; CV, cardiovascular.

**Table 4. Pharmacogenomic cohort population stratified by genotype and number of prescriptions.**

| | Amoxicillin (*n* = 37,497) | Clarithromycin (*n* = 7,121) | *p*-value |
|---|---|---|---|
| **rs1045642** | | | 0.29 |
| High P-gp (GG) | 7,995 (21.3) | 1,575 (22.1) | |
| Intermediate P-gp (GA) | 18,156 (48.4) | 3,433 (48.2) | |
| Low P-gp (AA) | 11,346 (30.3) | 2,113 (29.7) | |
| **rs1128503** | | | 0.15 |
| High P-gp (GG) | 11,516 (30.7) | 2,163 (30.3) | |
| Intermediate P-gp (GA) | 18,468 (49.3) | 3,599 (50.4) | |
| Low P-gp (AA) | 7,513 (20.0) | 1,373 (19.3) | |

*Allele dependent on genotyping platform used. P-gp, permeability-glycoprotein.

whereas in those who were not taking P-gp inhibitors, there was no significant increase in CV hospitalization with use of clarithromycin versus amoxicillin (HR 1.43; 95% CI 0.90–2.26, interaction *p*-value 0.029). This interaction was not seen at 15–30 days or 30 days to 1 year (15–30 days: P-gp, HR 0.93, 95% CI 0.49–1.76; no P-gp, HR 0.98, 95% CI 0.56–1.71, interaction p value 0.74; 30 days–1 year: P-gp, HR 0.95, 95% CI 0.73–1.25; no P-gp, HR 0.99, 0.77–1.28, interaction *p*-value 0.53).

## Pharmacogenomic cohort study

**Baseline characteristics.** In total, there were 37,497 amoxicillin prescriptions from 8,513 unique individuals and 7,121 clarithromycin prescriptions from 5,031 unique individuals. Patients who were prescribed clarithromycin were older (66.4 ± 12.4, [mean ± standard deviation] versus 63.2 ± 13.2 versus years, SMD 0.246) and more likely to be male (3,959/7,121 prescriptions [55.6%] versus 17,549/37,497 [46.8%], SMD 0.177). Individuals prescribed clarithromycin were also more likely to have had a prior MI (933/7,121 [13.1%] versus 1,687/37,497 [4.5%], SMD 0.307) and a history of COPD (2,135/7,121 [29.9%] versus 6,712/37,497 [17.9%], SMD 0.284). There was no significant difference in the percentage of patients with diabetes (6,472/7,121 [90.9%] versus 33,822/37,497 [90.2%], SMD 0.024).

The numbers of prescriptions and unique individuals available for analysis stratified by genotype are summarized in Table 4. The heterozygous (intermediate genetically predicted P-gp activity) phenotype was most common for both SNPs. There were no significant differences in prescribing of amoxicillin versus clarithromycin based on genotype.

## Association between clarithromycin use and CV hospitalization stratified by genetically predicted P-gp activity

Overall outcomes are summarized in Table 5. In total, there were 952 CV hospitalizations within 1 year of antibiotic prescription in the clarithromycin group (13.3% of all clarithromycin prescriptions) compared with 3,160 (8.4%) in the amoxicillin group. In this cohort, irrespective of genotype, clarithromycin prescription was associated with increased risk CV hospitalization compared with amoxicillin at 15–30 days and 30 days to 1 year (crude HRs: 0–14 days 1.88; 95% CI 1.48–2.38, *p* < 0.001, 15–30 days 2.27; 95% CI 1.74–2.95, *p* < 0.001, 30 days–1 year 1.57; 95% CI 1.45–1.70, *p* < 0.001; adjusted HRs: 0–14 days 1.24; 95% CI 0.96–1.60, *p* = 0.10, 15–30 days 1.50; 95% CI 1.14–1.99, *p* = 0.004, 30 days–1 year 1.10; 95% CI 1.01–1.19, *p* = 0.027).

**Table 5. Cardiovascular events in the pharmacogenomic cohort study.**

|  | Total Number of Prescriptions | CV Hospitalization | | |
|---|---|---|---|---|
|  |  | 0–14 days | 15–30 days | 30 days to 1 year |
| Amoxicillin | Total Number of Prescriptions | 37,497 | 37,238 | 37,053 |
|  | Number of Events (%) | 259 (0.7) | 185 (0.5) | 2,716 (7.3) |
| Clarithromycin | Total Number of Prescriptions | 7,121 | 7,030 | 6,951 |
|  | Number of Events (%) | 91 (1.3) | 79 (1.1) | 782 (11.3) |

When stratified by genotype, there was a significant gene-clarithromycin interaction for risk of CV hospitalization at 30 days to 1 year. Individuals who were homozygous for the allele associated with lower P-gp levels (AA) had significantly increased risk of CV hospitalization between 30 days and 1 year when prescribed clarithromycin compared with amoxicillin (rs1045642 AA: HR 1.39, 95% CI 1.20–1.60, $p < 0.001$, GG/GA: HR 0.99, 95% CI 0.89–1.10, $p = 0.85$, interaction $p$-value $< 0.001$ and rs1128503 AA 1.41, 95% CI 1.18–1.70, $p < 0.001$, GG/GA: HR 1.04, 95% CI 0.95–1.14, $p = 0.43$, interaction $p$-value $< 0.001$) (Table 6).

Restricting the analysis only to those individuals prescribed clarithromycin, for both the rs1045642 and rs1128503 genetic variants, individuals with the AA genotype were more likely to have a CV hospitalization between 30 days and 1 year than those with the GA or GG genotypes (rs1045642: HR 1.34, 95% CI 1.15–1.56, $p < 0.001$; rs1128503: HR 1.22, 95% CI 1.02–1.45, $p = 0.025$) (S2 Table).

## Discussion

We performed a cohort study in 2 parts, combining a traditional epidemiological approach with a pharmacogenomic study to evaluate the association of the use of the macrolide

**Table 6. Association of clarithromycin and cardiovascular hospitalization (versus amoxicillin) stratified by P-glycoprotein genotype.**

| SNP |  |  | rs1045642 |  |  |  |  | rs1128503 |  |  |
|---|---|---|---|---|---|---|---|---|---|---|
|  | Number of events (%) | Crude Hazard Ratio | Adjusted Hazard Ratio | p-value | Interaction p-value | Number of events (%) | Crude Hazard Ratio | Adjusted Hazard Ratio | p-value | Interaction p-value |
| **0–14 days** |  |  |  |  | 0.49 |  |  |  |  | 0.66 |
| GG/GA | 243 (7.8) | 2.07 (1.56–2.73) | 1.34 (0.99–1.82) | 0.06 |  | 280 (7.8) | 1.97 (1.52–2.57) | 1.31 (0.99–1.74) | 0.06 |  |
| AA | 107 (8.0) | 1.47 (0.93–2.34) | 1.03 (0.63–1.67) | 0.92 |  | 70 (7.9) | 1.50 (0.85–2.65) | 1.00 (0.54–1.84) | 0.99 |  |
| **15–30 days** |  |  |  |  | 0.51 |  |  |  |  | 0.11 |
| GG/GA | 206 (6.7) | 2.44 (1.82–3.27) | 1.54 (1.12–2.11) | 0.008 |  | 234 (6.6) | 2.11 (1.59–2.81) | 1.36 (1.00–1.84) | 0.049 |  |
| AA | 58 (4.3) | 1.72 (0.94–3.13) | 1.29 (0.69–2.42) | 0.42 |  | 80 (3.4) | 3.67 (1.77–7.63) | 2.77 (1.29–5.97) | 0.009 |  |
| **30 days–1 year** |  |  |  |  | <0.001 |  |  |  |  | <0.001 |
| GG/GA | 2,435 (7.9) | 1.47 (1.33–1.62) | 0.99 (0.89–1.10) | 0.85 |  | 2,830 (8.0) | 1.50 (1.37–1.64) | 1.04 (0.95–1.14) | 0.43 |  |
| AA | 1,062 (8.0) | 1.81 (1.58–2.08) | 1.39 (1.20–1.60) | <0.001 |  | 667 (7.6) | 1.88 (1.58–2.24) | 1.41 (1.18–1.70) | <0.001 |  |

AA, lowest genetically predicted P-glycoprotein levels; GA, intermediate genotype; GG, highest genetically predicted P-glycoprotein levels; SNP, single-nucleotide polymorphism. Adjusted hazard ratio adjusted for age at prescription, sex, history of type 2 diabetes, myocardial infarction, and chronic obstructive pulmonary disease.

antibiotic clarithromycin with CV risk. Our study has 2 key findings. First, not only was clarithromycin use associated with increased CV risk compared with amoxicillin, as other studies have reported, but we have identified that the risk is particularly increased in those taking P-gp inhibitors concurrently. This finding was strengthened by use of propensity-score weighting to adjust the results, and a similar finding was observed in the observational analysis of the genomic cohort. Second, in order to strengthen our findings, we have shown in a genomic study (which should reduce the risk of confounding by indication) that the association of clarithromycin with CV hospitalization between 30 days and 1 year was significantly increased in individuals with lower genetically predicted levels of P-gp activity. To the best of our knowledge, this is the first time that such an approach has been used to examine this subject. These results may suggest that, at least in part, the association of clarithromycin with increased CV risk may be modified via P-gp and particular caution may be needed when prescribing clarithromycin in individuals taking P-gp inhibitors.

## What this study adds to existing research

To the best of our knowledge, our study is the first to specifically examine a treatment interaction with P-gp inhibitors and substrates. Although clarithromycin is commonly recognized as a P-gp inhibitor [18], it is also a substrate for P-gp, and intracellular levels are increased when another P-gp inhibitor is co-administered [34, 35]. Supporting this, in our genomic analysis, we found that individuals with lower genetically predicted P-gp activity had a higher risk of CV hospitalization between 30 days and 1 year of clarithromycin prescription. Because of the random allocation of genotype at birth, pharmacogenomic studies are less likely to be affected by indication bias than traditional observational studies [36]. In a traditional observational study, this might account for an increase in CV risk seen with P-gp use. There was a higher crude number of events in the amoxicillin group than those prescribed clarithromycin (Table 2); however, after propensity-score adjustment, clarithromycin prescription was associated with worse outcome. This has been previously reported and is because patients prescribed amoxicillin alone tend to be older and more unwell; hence, methods such as propensity-score weighting are required to account for this [37], and this is likely to contribute to the nonsignificant unadjusted increase in mortality seen with amoxicillin in our study.

Concerns regarding the CV risk of macrolide antibiotics have been present for several years [38] and were strengthened by the results of the CLARICOR randomized trial, which, contrary to the authors' original hypothesis, demonstrated an increased risk of CV mortality at both 3 and 10 years [4, 16]. These results have been further supported by several large observational studies [5–7, 39] and meta-analyses [9, 10], which have reported increased CV risk of myocardial infarction and CV hospitalization up to 1 year after macrolide prescription. Other macrolides such as azithromycin have also been linked with increased CV risk [8]. Nevertheless, there have been alternative large observational studies that have suggested that there is no significantly increased CV risk with macrolide use [13, 14]. These alternative results suggest that there may be specific patient groups who are at particular risk when prescribed clarithromycin. The mechanism of increased CV risk with clarithromycin is not completely clear. The short-term increased risk of sudden cardiac death has been attributed to the effect of macrolides on the QT interval leading to arrhythmia, but this does not explain the increased long-term CV risk observed in CLARICOR and other studies that persisted after drug discontinuation. Alternative theories include macrophage activation leading to coronary plaque destabilization and acute coronary syndrome [40]. With these proposed mechanisms, any pathway by which metabolism of clarithromycin is impaired might lead to increased CV risk. Most studies have so far focused on the CYP3A4 enzyme; however, the interaction with macrolides has not been

consistent [6, 35]. We also did not find a significant interaction between clarithromycin and CYP3A4 inhibition.

## Strengths and limitations

The key strength of our study is our use of both inverse probability of treatment weighting in our longitudinal cohort and a pharmacogenomic study. By using both of these methods, we strengthen the evidence supporting our finding that the increase in CV risk following clarithromycin prescription is associated with P-gp. Our study has some limitations. First, there are inherent limitations with any observational study, although our use of propensity weighting for likelihood of prescription and genomics do obviate some of these. Nevertheless, it is possible that unmeasured confounding could affect our results. Second, our pharmacogenomic cohort was mainly white, and further studies are required to determine whether our findings in this group are applicable to other ethnicities. As there have been no large genome-wide association studies looking at P-gp activity, we were unable to construct a weighted genetic risk score to evaluate the cumulative effect of P-gp SNPs. A large genome-wide association study would provide more precision around the effect of genetic variants. We used electronic health records and ICD coding to determine prescribing and outcomes; thus, we could not determine adherence. We could also not robustly ascertain prescribing indication and adjust for this, although the majority of prescriptions are likely to have been for suspected lower respiratory tract infection. Nevertheless, current coding accuracy in Scotland is considered reliable [41–43]. Furthermore, any inconsistency would affect both clarithromycin and amoxicillin cohorts equally. We used amoxicillin as a comparator as it is the most-commonly prescribed antibiotic for this indication in the UK, and we did not evaluate other antibiotic classes. We did not have any allergy information, which might also provide another source of indication bias.

## Clinical implications and next steps

Macrolides are primarily used to treat respiratory infections; however, alternatives such as amoxicillin or tetracyclines are not associated with increased CV risk, are not metabolized through P-glycoprotein, and have been shown in randomized trials to be noninferior to treatment regimens including macrolides [44, 45]. The most recent American Thoracic Society guidelines for community-acquired pneumonia recommend the use of macrolide monotherapy only in areas where macrolide resistance is low and in patients in whom alternative antibiotics are contraindicated [46]. Given that drugs such as calcium channel blockers are widely used (in 1 cohort, concurrently prescribed in up to 20% of patients with respiratory infections) [8] and such patients may already be at increased CV risk because of their underlying drug indication, advice to be cautious with clarithromycin seems justified as there are equally efficacious alternatives that do not carry this increased risk. This is consistent with advice from the US FDA that proposes caution with clarithromycin use in those with increased CV risk. Our study heralds the possibility of "precision" prescribing, in which patients are prescribed alternative antibiotics if they are taking P-gp inhibitors or if they have a particular genotype. Recent work has suggested the potential for P-gp-associated SNPs to be used in pharmacogenomic strategies for prescribing in other settings [47]. Mechanistic studies evaluating the pathophysiology of P-gp-associated gene and drug interactions in detail would further our understanding and inform future clinical practice.

## Conclusion

We found that clarithromycin use was associated with an increased risk of CV hospitalization up to 1 year post-prescription compared with amoxicillin. There appears to be an effect

modification via P-gp, with a particularly increased risk of adverse CV events with clarithromycin in patients also taking drugs that are P-gp substrates or those with lower genetically predicted levels of P-gp activity.

## Supporting information

**S1 Checklist. STROBE Checklist.** STROBE, Strengthening the Reporting of Observational Studies in Epidemiology.
(DOCX)

**S1 Text. Initial statistical analysis plan submitted at the time of funding application.**
(DOCX)

**S1 Table. Subgroup analysis of unadjusted hazard ratios for the association of clarithromycin with cardiovascular hospitalization at 14 days in the longitudinal cohort study.**
(DOCX)

**S2 Table. Association of AA genotype (lowest genetically predicted P-glycoprotein levels) with CV hospitalization compared with other GG or GA genotype in patients prescribed clarithromycin.** CV, cardiovascular.
(DOCX)

## Acknowledgments

We wish to thank Dr. Mike Lonergan for his help with statistical revision of the manuscript.

## Author Contributions

**Conceptualization:** Ify R. Mordi, Chim C. Lang, James D. Chalmers.

**Data curation:** Ify R. Mordi, Colin N. A. Palmer.

**Formal analysis:** Ify R. Mordi, Benjamin K. Chan.

**Funding acquisition:** Ify R. Mordi, Chim C. Lang, James D. Chalmers.

**Investigation:** Ify R. Mordi.

**Methodology:** Ify R. Mordi, N. David Yanez.

**Project administration:** Ify R. Mordi.

**Resources:** Ify R. Mordi.

**Supervision:** Chim C. Lang, James D. Chalmers.

**Writing – original draft:** Ify R. Mordi, Benjamin K. Chan, James D. Chalmers.

**Writing – review & editing:** Ify R. Mordi, Benjamin K. Chan, N. David Yanez, Colin N. A. Palmer, Chim C. Lang, James D. Chalmers.

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
