## [Decision Letter · Decision Letter 0]

13 May 2020

Dear Dr. Mordi,

Thank you very much for submitting your manuscript "Drug Interactions Through P-Glycoprotein Increase Cardiovascular Risk Associated with Clarithromycin: An Epidemiological and Genomic Population-Based Cohort Study" (PMEDICINE-D-19-03443) for consideration at PLOS Medicine. 

We apologize for the lengthy review process, and appreciate your patience. Your paper was evaluated by a senior editor and discussed among all the editors here. It was also sent to two independent reviewers, including a statistical reviewer. The reviews are appended at the bottom of this email and any accompanying reviewer attachments can be seen via the link below:

[LINK]

In light of these reviews, I am afraid that we will not be able to accept the manuscript for publication in the journal in its current form, but we would like to consider a revised version that addresses the reviewers' and editors' comments. Obviously we cannot make any decision about publication until we have seen the revised manuscript and your response, and we plan to seek re-review by one or more of the reviewers. 

We expect to receive your revised manuscript by Jun 03 2020 11:59PM. Please email us (plosmedicine@plos.org) if you have any questions or concerns.

We look forward to receiving your revised manuscript. 

Sincerely,

Caitlin Moyer, Ph.D.

Associate Editor 

PLOS Medicine

plosmedicine.org

1.Title: Please revise the title to avoid implications of causality, and also indicate the population/setting in the title, we suggest: “Genetic and pharmacological relationship between P-Glycoprotein and increased cardiovascular risk associated with clarithromycin prescription: An Epidemiological and Genomic Population-Based Cohort Study in Scotland, UK” or similar.

2. Prospective Analysis Plan: Did your study have a prospective protocol or analysis plan? Please state this (either way) early in the Methods section.

c) In either case, changes in the analysis—including those made in response to peer review comments—should be identified as such in the Methods section of the paper, with rationale

3. Data Availabilty Statement: PLOS Medicine requires that the de-identified data underlying the specific results in a published article be made available, without restrictions on access, in a public repository or as Supporting Information at the time of article publication, provided it is legal and ethical to do so. Please see the policy at 

http://journals.plos.org/plosmedicine/s/data-availability

and FAQs at 

http://journals.plos.org/plosmedicine/s/data-availability#loc-faqs-for-data-policy

Additionally, please provide a link or file containing the STATA and R code as requested by reviewer 1.

4. Abstract: Please provide some of the relevant summary demographics for the observational cohort study in Scotland. Please include some information on the population and years during which the study took place for the GoDARTS participants. 

5. Abstract: Please quantify the main results with both 95% CIs and p values. Where applicable, please include the important dependent variables that are adjusted for in the analyses.

6. Abstract: In the last sentence of the Abstract Methods and Findings section, please describe the main limitation(s) of the study's methodology.

7. Author Summary: At this stage, we ask that you include a short, non-technical Author Summary of your research to make findings accessible to a wide audience that includes both scientists and non-scientists. The Author Summary should immediately follow the Abstract in your revised manuscript. This text is subject to editorial change and should be distinct from the scientific abstract.

Please see our author guidelines for more information: https://journals.plos.org/plosmedicine/s/revising-your-manuscript#loc-author-summary

8. Results: Please present p values as p<0.001 where applicable.

9. Results: At the bottom of page 9: Please revise the following sentence, as the term "trend" is used to refer to a nonsignificant p value. The term trend should be used only when the test for trend has been conducted. “There was also a non-significant trend to higher likelihood of hospitalisation for MI within the first 14 days of clarithromycin.”

10. Results: Please present the numerators and denominators when reporting percentages (at least in a table if not in the text); for example, describing the pharmacogenetic cohort: “Individuals prescribed clarithromycin were also more likely to have had a prior MI (13.1% vs. 4.5%, p<0.001) and a history of COPD (29.9% vs. 17.9%, p<0.001).”

11. Results: Please present both the unadjusted and adjusted results where applicable, at least in the tables if not in the text. For example, please present the unadjusted results for the relationship between clarithromycin and CV hospitalization risk described on page 13.

12. Results: Page 13: Please clarify this sentence, as it seems like there was also an increased risk at 15-30 days. “After adjustment for age, sex, history of myocardial infarction and history of COPD, clarithromycin prescription was associated with increased risk of CV hospitalisation between 30 days and 1 year (0-14 days: HR 1.23, 95% CI 0.95-1.58, p=0.12; 15-30 days: HR 1.50, 95% CI 1.13-1.99, p=0.005; 30 days-1 year: HR 1.10, 95% CI 1.01-1.19, p=0.031).” 

13. Discussion, page 15: Please temper the following sentence with “To the best of our knowledge” or similar: “Our study is the first to specifically examine a treatment interaction with P-glycoprotein inhibitors and substrates.”

14. Discussion: Please present and organize the Discussion as follows: a short, clear summary of the article's findings; what the study adds to existing research and where and why the results may differ from previous research; strengths and limitations of the study; implications and next steps for research, clinical practice, and/or public policy; one-paragraph conclusion. You touch on implications for clinical practice throughout the discussion, but please consolidate together in a paragraph if possible.

15. Discussion (and throughout): Please replace "Caucasian" with "white" throughout the paper.

16. Table 1: For age, please indicate in the table or legend if you are reporting mean +/- SD, or some other measure here.

17. Table 2: Please clarify what the values are representing in the legend “Figure in brackets refers to the number of events as a percentage of the total number of prescriptions.” as there are parentheses in the table but no brackets. If the values in parentheses are the percentages of the total number of prescriptions, please place the number of prescriptions in the column headers, for reference. In the legend, please define the abbreviation for ‘MI’.

18. Table 3: Please present the unadjusted results in addition to the adjusted HRs. Please define abbreviations for MI, CI, and IPTW in the legend.

19. Figures 1 and 2: Please present the unadjusted results as well. Please specify the variables controlled for in the legend, and define abbreviations for CV, MI, COPD, HF, PGP, CCB, and CI.

20. References: Please place in-text citations in square brackets, like this [1].

21. Checklist: Thank you for including the STROBE checklist as Supporting Information. Please revise the checklist, using section and paragraph numbers, rather than page numbers to refer to locations in the manuscript. 

Comments from the reviewers:

Reviewer #1: The paper examines the effect of possible drug-drug interactions between clarithromycin and P-glycoprotein (P-gp) inhibitors on cardiovascular risk. The authors consider one observational and one pharmacogenomic cohort in a setting that contrasts how prescriptions of clarithromycin and amoxicillin impact future hospitalizations due to cardiovascular events. The results highlight a P-gp-centric molecular mechanism explaining the bioavailability of clarithromycin and suggest the existence of an at-risk population with a genetic variant coding for low P-gp activity.

Overall, the paper is well-written and easy to follow. The hypothesis is clearly formulated and placed in the context of published literature. The usage of observational and pharmacogenomic cohorts strengthens the findings, which, if confirmed, may have an immediate impact on the prescription and dosage of clarithromycin in the clinic. However, several aspects of the statistical analysis currently give pause and must be revisited to ensure that the findings and conclusions are robust.

Since the focus is on drug-drug interactions, the authors need to be more clear about when clarithromycin is considered by itself vs. in combination with P-gp inhibitors. The Methods section states that the study contrasts "all patients over 18 years old who were prescribed clarithromycin (alone or in combination with another antibiotic) over this period, with those prescribed amoxicillin only as a control group", which gives the impression that clarithromycin monotherapy and combination therapy are always considered as a single group. If this is the case, then how do we know that the clinical outcomes reported in Tables 2 and 3 are not confounded by whether clarithromycin was prescribed alone or in combination with a P-gp inhibitor since amoxicillin is always considered as a monotherapy? (In fact, the subsequent analysis reported in Figures 1 and 2 shows that such a confounder is indeed likely.)

Along the same vein, I am not sure if amoxicillin monotherapy is a proper control for evaluating the effect of prescribing clarithromycin in combination with a P-gp inhibitor. It seems that either 1) amoxicillin should also be paired with a P-gp inhibitor, or 2) the control group should consist of concomitant prescriptions of clarithromycin and "another drug that is not a P-gp inhibitor". Otherwise, one could argue that all the observed differences in clinical outcomes are simply due to the number of prescribed drugs.

The pharmacogenomic analysis is well formulated and cleanly executed, other than the possible ramifications of the control group considerations above.

All reported p-values need to be adjusted for multiple hypothesis testing. The authors state that "a p value <0.05 was considered statistically significant", but without adjustment this significance threshold implies that approximately one out of every 20 tests may produce a false positive.

Minor comments:

1. The statement "To the best of our knowledge, there have been no studies evaluating the association between clarithromycin use, CV risk and P-gp." is a bit too strong. The study by Wessler and colleagues (which is cited in the manuscript) has certainly considered clarithromycin in the context of CV risk and P-gp, as part of a larger study.

2. The STATA and R code used for the analysis should be released alongside the manuscript (e.g., as a GitHub repository) to increase reproducibility.

3. What is the purpose of reporting a p value for "Total Number of Unique Patients" and "Total Number of Prescriptions" in Table 1? Is the goal to demonstrate that overall one drug is prescribed significantly more often than the other? I don't think that's very surprising...

Reviewer #2: This is an interesting analysis of an important but poorly understood phenomenon, the long-term cardiovascular risks following use of clarithromycin. I believe it would be a useful addition to the literature but there are several ways the manuscript could be improved and there are also some items that need to be clarified. 

My major concern with the longitudinal cohort study is the possibility of residual confounding; in particular, indication was not accounted for, a limitation which should be noted in the Discussion. The only two significant HRs are modest in magnitude (1.31, 1.13) and so are in the range that could be the result of confounding. Were the p-values in Table 3 adjusted for multiple comparisons? Also, when you discuss the impact of weighting in the Discussion you note that before weighting the risk of CV events was higher with amoxicillin. It wasn't clear what result that refers to; was it the 15-30 day window? 

The most persuasive evidence for a role of P-gp comes from the pharmacogenomic study and the interaction seen with genetically determined low P-gp activity, which as point out is unlikely to be due to confounding. I think you should seriously consider revising the paper to highlight the pharmacogenomic study rather than the longitudinal cohort study, perhaps by switching the order in which the studies are described in the manuscript. Have you compared the risk within clarithromycin-exposed patients alone; i.e., clarithromycin users with GG/GA genotype versus clarithromycin users with AA genotype? This comparison, if significant, would be even more persuasive, but may not be adequately powered in your sample, however.

Regarding Table 1 (observational longitudinal cohort study): Use of a t-test or chi-square test to assess baseline differences between groups results in many statistically significant differences which are small in magnitude, but statistically significant because of the large sample size. A better approach would be to compare the groups with standardized differences, for which a difference of 0.1 or more is considered consequential. After IPTW, the standardized differences can be compared again to determine if the weighting successfully balanced the baseline covariates. Your manuscript does not address the issue of whether the IPTW succeeded in balancing the baseline covariates. See Mamdani et al., at https://www.bmj.com/content/330/7497/60.long

In the observational longitudinal cohort study, it is puzzling why the mean age is so high (about 73 years for clarithromycin users), if any patient over 18 years of age was included. Can you please comment? 

Study endpoints: Was ICD-10 in use in Scotland during the entire study period (2004-2014)? 

Minor comments

Table 2: suggest showing total number of patients per group in the header row

Figure 2: is the risk window here also 14 days? Are all these drugs P-gp substrates? If so, why does there only appear to be an interaction with dihydropyridine CCBs and maybe amiodarone? 

Table 4: the percentages for rs1045642 don't sum properly (total = 110.5%)

Discussion: You state that amoxicillin and tetracyclines are non-inferior to macrolides for respiratory infections. Some have argued that macrolides reduce mortality from community acquired pneumonia, however. See for example Asadi et al., https://academic.oup.com/cid/article/55/3/371/613119

Final sentence: It is probably more appropriate to speak of drugs that are substrates for P-gp rather than "metabolized" by P-gp.

[LINK]

---

## [Decision Letter · Decision Letter 1]

13 Aug 2020

Dear Dr. Mordi,

Thank you very much for re-submitting your manuscript "Genetic and pharmacological relationship between P-Glycoprotein and increased cardiovascular risk associated with clarithromycin prescription: An Epidemiological and Genomic Population-Based Cohort Study in Scotland, UK" (PMEDICINE-D-19-03443R1) for review by PLOS Medicine.

I have discussed the paper with my colleagues and the academic editor and it was also seen again by two reviewers. I am pleased to say that provided the remaining editorial and production issues are dealt with we are planning to accept the paper for publication in the journal.

[LINK]

We look forward to receiving the revised manuscript by Aug 20 2020 11:59PM. 

Sincerely,

Caitlin Moyer, Ph.D.

Associate Editor 

PLOS Medicine

plosmedicine.org

Requests from Editors:

1. Data availability statement: As mentioned by reviewer 1, please include the GitHub link specific for your dataset used in this study.

2. Abstract: Background: First sentence: Please change the word “and” to “with”

3. Abstract: Background: Please revise to temper this claim, we suggest: “To the best of our knowledge, no studies have examined whether this association might be mediated via P-glycoprotein (P-gp), a major pathway for clarithromycin metabolism.”

4. Abstract: Methods and Findings: There appears to be a missing p value for the pharmacognetic report of the AA allele associated with lower p-gp activity (rs1045642 AA: HR 1.39, 95% CI 1.20-1.60)

5. Abstract: Conclusions: We suggest revising to: “In this study, we observed that that increased risk of CV events with clarithromycin compared to amoxicillin were associated with an interaction with P-glycoprotein.” or similar to temper the causal implications.

6. Author summary: What did the researchers do and find?: We suggest combining the third and fourth bullet points as follows:

“-In this analysis we found that that patients prescribed clarithromycin were significantly more likely to have a cardiovascular hospitalisation at 0-14 days and 30 days to 1 year after prescription than those prescribed amoxicillin, and that individuals who were co-prescribed P-glycoprotein substrates or inhibitors and clarithromycin had significantly higher risk of cardiovascular hospitalisation.”

7. Author summary: What do these findings mean: We suggest revising the second point as follows:

“These results suggest implications for clarithromycin use patients taking P-glycoprotein inhibitors or with low genetically-predicted P-glycoprotein activity.” or similar.

8. Data availability statement: Please remove this section from the body of the manuscript and ensure it is entered accurately (with updated GitHub information) where appropriate in the manuscript submission system form.

9. Methods: Page 7: Regarding your prospective analysis plan, thank you for noting in your response to comments that “We did not have a pre-specified plan for the pharmacogenomic section of the study, however, the analysis was informed by the cohort study.” Please include a statement such as this in Methods.

10. Methods: Study Endpoints: First sentence, and throughout: Please consistently use the abbreviation “CV” for cardiovascular, defining the abbreviation at first use in the text and using the abbreviation throughout, for the sake of consistency: “The primary endpoint for both studies was CV hospitalisation. In our 1 initial funding proposal we planned to evaluate cardiovascular mortality as the primary endpoint…”

11. Methods: Please add the following statement, or similar, to the Methods: "This study is reported as per the Strengthening the Reporting of Observational Studies in Epidemiology (STROBE) guideline (S1 Checklist)."

12. Results Page 11: Clinical outcomes for Observational Cohort Study: Please provide the result with 95% CIs and p values here: “A higher proportion of those taking clarithromycin had MI requiring hospitalisation within 14 days, though this difference was not statistically significant.”

13. Results: Page 11: Please also present the HR and 95% CIs for the interactions with P-gp inhibitors at 30 days and 1 year to accompany these p values? “interaction was not seen at 30 days or 1 year (interaction p values 0.74 and 0.53 respectively).”

14. Discussion: Please start the discussion with 1-2 sentences briefly summarizing what was done in the study.

15. Discussion: Page 13: Please revise to: “These results may suggest that, at least in part, the association of clarithromycin with increased CV risk may be modified via P-glycoprotein and particular caution may be needed in prescribing clarithromycin in individuals taking P-gp inhibitors” or similar

16. Discussion: Page 13: Please revise to: “This has been previously reported and is because patients prescribed amoxicillin alone tend to be older and more unwell, hence methods such as propensity-score weighting are required to account for this [37], and this is likely to contribute to the non-significant unadjusted increase in mortality seen with amoxicillin in our study” or similar.

17. Discussion: page 14: Please change “was” to “is” in the following sentence: “With these proposed mechanisms, any pathway by which metabolism of clarithromycin was impaired might lead to increased CV risk.”

18. Discussion: page 15: Please replace the term “compliance” with “adherence” where it is used to refer to treatment adherence.

19. Discussion: Page 15: Please remove the word “which” to clarify: “The most recent American Thoracic Society guidelines for community acquired pneumonia recommend the use of macrolide monotherapy only in areas where macrolide resistance is low and in patients in whom alternative antibiotics are contraindicated [46].”

20. Sections: Declaration of Interests, Funding, Role of the Funding Source: Please remove these sections from the main text of the manuscript, and ensure the information is accurately entered into the relevant locations within the manuscript submission system.

21. Table 1: Please indicate in the table or legend that the values represent numbers and percentages.

22. Table 3: Please define the abbreviation “CI” in the legend.

23. Table 6: Please also define GA/GG in the legend. Please indicate in the column headers that the hazard ratios are followed by the 95% CIs.

24. Figure 1 and Figure 2: Please also define abbreviation for CV in the legend.

25. S1 Table: Please define abbreviations for pgp, CCB, and CI in the legend.

26. Supplementary File 2: Analysis Plan: Thank you for including your analysis plan. If you have a dated version of this plan, please update accordingly.

Comments from Reviewers:

Reviewer #1: Overall, I am satisfied with the revisions submitted by the authors and have no other concerns. In particular, I thank the authors for clarifying the nuances of clarithromycin prescription. With the new description, I no longer have reservations about the control group. Very minor comments are below.

I agree with the authors that the Bonferroni correction is too stringent and would generally recommend something like the recently-proposed harmonic mean p-value (PMID: 30610179), which is able to handle groups of dependent tests without explicit access to what the dependency structure is. However, I think that reporting raw p-values without placing an artificial "significance" threshold is also a reasonable strategy, albeit one that softens the main message.

Minor note regarding GitHub organization: The GitHub link in the manuscript (https://github.com/ifymordi) refers to a user, not a repository. As this user (I hope) publishes additional code in future studies, it will become increasingly difficult to find the code associated with the current clarithromycin study by following the link in the paper. Consider renaming your "Research" repository to "Clarithromycin" and updating the link to github.com/ifymordi/Clarithromycin. It also seems that all code is in a branch that is not immediately visible to a user. Consider merging this branch into master, since that's what readers will see when they first follow your link.

Reviewer #2: I appreciate the opportunity to review the revised manuscript. I thank the authors for their careful and complete consideration of my comments. The manuscript is much improved and while I have a couple of suggestions I don't feel that any further changes are mandatory. However, for the authors' consideration, I will provide some further thoughts regarding two of my previous comments.

1. Standardized differences--I would point out that it is possible to calculate standardized differences in baseline characteristics on the weighted population, just as you have now done for before IPTW. This is similar to calculating standardized differences on a propensity-score matched sample both before and after matching. If you show that the post-IPTW standardized differences are minimal (<0.1) that would strengthen the findings of the study. 

2. Regarding my suggestion for a within-clarithromycin analysis of risk by P-gp genotype: I take the point that the test for the treatment X genotype interaction essentially controls for the effect of amoxicillin. However, I wonder if a within-clarithromycin analysis might provide a more straightforward demonstration of this effect, although one would have to assume that P-gp genotype varies randomly among clarithromycin users. This is just a suggestion for the authors as the existing analysis does address the effect of genotype.

[LINK]

---

## [Editor Report · Decision Letter 2]

21 Sep 2020

Dear Dr. Mordi, 

On behalf of my colleagues and the academic editor, Dr. Sanjay Basu, I am delighted to inform you that your manuscript entitled "Genetic and pharmacological relationship between P-Glycoprotein and increased cardiovascular risk associated with clarithromycin prescription: An Epidemiological and Genomic Population-Based Cohort Study in Scotland, UK" (PMEDICINE-D-19-03443R2) has been accepted for publication in PLOS Medicine. 

PRODUCTION PROCESS

PRESS

PROFILE INFORMATION

Thank you again for submitting the manuscript to PLOS Medicine. We look forward to publishing it. 

Best wishes, 

Caitlin Moyer, Ph.D.

Associate Editor 

PLOS Medicine

plosmedicine.org